# Personal data in EU digital wallets in light of the recent developments in EU data protection law

Bilgesu Sumer [1,*,†]

[1]*KU Leuven (Belgium), Sint Michielsstraat 6, Leuven, 3000, Belgium*

**Abstract**

This paper discusses the proposed changes to the definition of personal data under the GDPR by the Digital Omnibus Proposal (DOP). It applies this new interpretation to the EU Digital Wallet ecosystem to assess whether the DOP would achieve its promises, namely bringing simplification and legal clarity. It essentially argues that the proposed changes might not bring clarity, and on the contrary, risk compounding the existing uncertainties and legal risks.

**Keywords**

GDPR, Digital Wallets, SRB, Personal data, Controller

## 1. Introduction

Digital identity wallets (DIWs) have emerged as a central infrastructure in decentralised identity management systems, often also referred to as a key component of the self-sovereign identity (SSI) ideal, enabling individuals to manage their identifiers and credentials in a user-centric fashion [1]. Digital wallets primarily serve as repositories for cryptographic keys and identity-related artifacts that enable people to authenticate themselves and selectively disclose attributes or claims to third parties[2][3].The demand for identity management has shifted toward "self-sovereign app-based' wallets that are managed through the user's mobile device under the user's 'full control' [4]. Accordingly, recent regulatory frameworks, e.g., the European Digital Identity Framework (EUDIF) – also known as eIDAS 2.0, and industry implementations focused on decentralising identity management systems tend to centre on digital wallets[5].

The prevailing view on whether DIWs process personal data – as defined in the GDPR ('any information relating to an identifiable natural person') – has so far been affirmative. For example, the European Data Protection Supervisor (EDPS) describes DIWs as applications that allow the secure storage, management, and sharing of personal identification data and other personal attributes [6][7]. Accordingly, access to a DIW could reveal personal data [7].

Nevertheless, there is no specific academic source discussing the legal nature of the data in DIWs, an overview of the literature on personal data and decentralised technologies demonstrates that this question is not easy to answer.[8][9] The literature shows that, when applied to decentralised and user-centric data infrastructures, persistent questions arise regarding data classification and responsibility under the GDPR. The application of the GDPR is seen as unclear for two main reasons: (i) decentralised technologies, including personal data stores and digital wallets, challenge the traditional, centralised assumptions upon which the GDPR is built[8]; (ii) GDPR's expansive, context-dependent notion of "personal data": the concept has become so broad that it is referred to as the "law of everything.'[10] In particular, data access and pseudonymisation techniques—such as encryption—used in digital wallets and comparable decentralised technologies (e.g., personal data stores like Solid) have generated uncertainty as to both the scope of the GDPR and the allocation of responsibilities.[11]

*SoSy2026-Privacy: 4th Privacy Personal Data Management Session @ Solid Symposium 2026 , London, UK*

*Corresponding author.

✉ bilgesu.sumer@kuleuven.be (B. Sumer)

🌐 https://www.law.kuleuven.be/citip/en/staff-members/staff/00139995/ (B. Sumer)

🆔 0000-0003-0626-6726 (B. Sumer)

This contribution addresses the question of the extent to which recent developments in EU data protection law clarify the legal nature of personal data, if any, processed by the European Digital Identity Wallet (EUDI)? In doing so, I conduct a doctrinal legal analysis, based on the Court of Justice (CJEU) case law, relevant GDPR and EUDIF Regulation, and rely on the EUDIW Architecture Reference Framework. I rely on this framework to map data categories and actors, and to assess whether personal data are processed, and from whose perspective. [12].

## 2. Background: Uncertainty of the concept of personal data in decentralised contexts and digital wallets

The GDPR currently has a more objective approach to personal data. Article 4(1) defines personal data as any information relating to an identified or identifiable natural person (data subject). Recital 26 states: "To determine whether a natural person is identifiable, account should be taken of all the means reasonably likely to be used, such as singling out, either by the controller or by another person to identify the natural person directly or indirectly." Recital 30 of the GDPR states that:

> The presence of any unique identifiers is enough to make the data identifiable, and therefore not anonymous. Natural persons may be associated with online identifiers provided by their devices, applications, tools, and protocols, such as internet protocol addresses, cookie identifiers, or other identifiers such as radio frequency identification tags. This may leave traces which, in particular when combined with unique identifiers and other information received by the servers, may be used to create profiles of the natural persons and identify them

In the Breyer Case, the Court ruled that, for information to be classified as personal data (regarding dynamic IP addresses), it is not necessary that all the information enabling the identification of the data subject be held by a single person (para 43)[13]. To determine if a person is identifiable, consider all means that could reasonably be used by the controller or any other person to identify them (para 42)[13]. The entity had legal means of receiving additional data from another entity to identify the data subject. The Court established that data remains personal even if it requires legal means to identify an individual. This includes situations in which a processing entity may contact a competent authority to use data in a legal proceeding to identify relevant persons, e.g., after or during a cyber-attack (para 47)[13].

In Scania, the Court emphasised the possibility of indirect identifiability. A vehicle's registration number VIN constitutes personal data when someone, e.g., a repairer, has the means to associate it with a natural person. While for some entities VINs do not automatically constitute personal data, they become personal data indirectly when they are made available to others who have the means to identify the individual (para 49)[14].

In light of the above, there are two approaches to identifiability: (i) absolute and (ii)relative approaches. The former holds that a person is considered identifiable if anyone can identify them. Thus, the nature of the data itself is the parameter to determine whether the data in question is identifiable. In contrast, the latter considers a person identifiable only if the data controller itself has the means to identify them, relying solely on its own capacity. [15] While the GDPR provisions tend to align with the absolute approach to personal data, the CJEU's stance on this was leaning towards a relative approach, with a nuance: the data constitute personal data if it is identifiable by the controller or any other person, which essentially matched the GDPR wording in Recital 26. A consequence of this is that pseudonymised data would generally be considered personal data for any person who could access it and has the means to identify the natural person linked to that pseudonym. Consequently, the definition was highly broad to provide legal protection to data subjects.

It should be noted that, to the author's knowledge, analysis of personal data has not been conducted specifically for EUDIW. This type of analysis has been done for other types of decentralised technologies,

e.g., blockchains and personal data stores, which usually focus on an absolute approach and do not delve into an actor-specific analysis – which was understandable considering the explicit wording in the GDPR[16][11]. Applying the absolute approach above, if one piece of personal data qualifies as personal data for one party in the same chain of processing, it would mean that personal data exists in that chain, thereby affecting each entity's responsibilities within it. For example, while being essential for enhancing privacy, cryptographic selective disclosure techniques, particularly the use of zero-knowledge proofs (ZKPs), as one of the core functions of DIWs, would not eliminate the applicability of the GDPR, as some entities in the same processing chain could still identify individuals.[17]

## 3. EDPS VS SRB Case and the Changes Proposed in the Digital Omnibus Proposal

### 3.1. The SRB vs EDPS Case

The CJEU's EDPS vs SRB case were delivered on 4 September 2025 and had a ripple effect within the data protection community. The case examines whether pseudonymised shareholder comments collected during a bank resolution qualify as personal data when shared with a third-party audit firm, Deloitte. The Court ruled that personal opinions in the comments inherently relate to the individuals who express them. Because the SRB had the means to identify the commentators, the information was deemed personal data for the SRB, regardless of whether Deloitte could identify them [18].

The SRB case did not drastically change how personal data should be interpreted, but it introduced significant nuances regarding pseudonymisation and a stricter, relative approach to personal data. Pseudonymised data must not, in all cases, be considered personal data —it is case-dependent. In some cases, pseudonymisation can effectively prevent identification by other entities (other than the controller who conducts the pseudonymisation) (para 86)[18]. After a pseudonymisation technique is applied, the data remain personal for a controller who retains the additional identifying information. Moreover, the Court stated that identifiability should be assessed at the time data are collected, and that data subjects should be informed accordingly. The purpose of this obligation is to allow the data subject, in full knowledge of the facts, to decide whether to provide the personal data being collected (para 109)[18]. In summary, the Court held that the definition of personal data is intrinsically linked to the means available to both the controller and other parties to identify the natural person concerned (para 16)[18].

### 3.2. Digital Omnibus Proposal and the changes to the interpretation of personal data

The DOP introduces targeted amendments to the definition of personal data in Article 4(1) to clarify when information should be considered as relating to an identifiable person. In a way, it codifies the SRB decision, but takes a narrower view on personal data. According to this, the status of personal data should be relative to the entity holding it, i.e., relative identification, which is explained as:"information is not necessarily personal data for every entity merely because another entity can identify that natural person." The main parameter to understand whether the data in question is personal data to assess whether a specific entity has the means "reasonably likely to be used' to identify the individual (Article 3 (a))[19]. Importantly, data does not become personal for an entity just because a subsequent recipient might have the tools, e.g., other datasets to link, to identify the person – which clearly contradicts with the Scania case[14].

According to the Commission (Recital 27 DOP), the change assists controllers in assessing whether pseudonymised data genuinely qualifies as personal data within their particular context. To facilitate this, the Commission and the European Data Protection Board (EDPB) will create technical criteria and implement them through acts to define the latest methods for evaluating re-identification risks[19].

## 4. A legal analysis and discussion of the changes in the context of digital identity wallets

### 4.1. Data and Actor mapping

The EUDI Wallet Architecture and Reference Framework (ARF) outlines the structure, components, and interactions within the EUDI Wallet ecosystem.[12] Without being exhausted and purely based on the ARF, five main categories of data can be identified: (i) Personal identification data (PID); (ii) Electronic Attestation Attributes (EAA); (iii) Pseudonyms; (iv)Wallet Unit Attestation (WUA); (v) Operational and transactional data (transaction logs) [12].

PIDs are legally issued sets of data used to establish a person's identity. EAA are data models that verify specific attributes of a person – data is exchanged in the form of EEA(ARF 5.1).[12]. Pseudonyms are user-chosen identifiers used to authenticate to online services without revealing legal identity. WUAs are attestations issued by the wallet provider to verify the authenticity of wallets for other providers. Finally, the wallet maintains operational and transactional data within a transaction log accessible via a dashboard, which records the date, time, counterpart identification, and specific data shared to ensure user transparency and oversight of all past interactions.[12].

As mentioned, the novel approach to personal data hinges on the identifiability capabilities (both legal and practical) of the entities involved in the processing. From the ARF, five entities that process data through the wallet ecosystem can be identified as: (i) PID providers; (ii) attestation providers; (iii)wallet providers; (iv)relying parties and (v)trust service providers (TSPs) [12]. Another entity usually involved in academic discussions in decentralised processing is the data subjects as data controllers themselves[16]. I will not discuss this in the context of digital identity as it is not relevant to this topic or discussion.

In a typical interaction, i.e., an authentication and verification scenario involving wallets, a user (identity holder) displays data from their PID, which is issued and managed by PID providers, organisations mandated by the Member States. As stated in the ARF, they can be the same organisations that issue official identity documents or wallet providers, or both at the same time. These data are presented to various public and private online services (relying parties)(ARF 3.4). Relying Parties can ask for specific PID attributes in each transaction (3.11)[12].

### 4.2. Who might be processing personal data?

In light of the new developments introduced above, any analysis here should be done considering which actor has access which data and where it is pseudonymised, whether this entity has reasonably likely means to identify the users.

First of all, I will argue that both relying parties and providers of PIDs and EAAs typically have additional information about the user. Hence, the data processed by these parties and the data in the wallet unit may be considered personal data. Most Relying Parties already have a customer account, or they create one as part of their service, which can hold email addresses, account information, or payment information. The wallet presentation (attribute) can be only about the age of the user (as an obligation for some online platforms)[20][21]. However, as a practical reality, the wallet attestation can be linked to this account information.

PID providers, as inherent in their role, keep civil registry data, e.g., name, date of birth, nationality, identity numbers, etc. This means that any wallet-related identifier that is available to an issuer can be linked to an identified person. In other words, PID issuers possess means reasonably likely to be used for some of the data processed through EUDI.

While PID providers inherently process additional identifying data in conjunction with their authoritative legal identity records, private-sector EAA providers might process such data alongside individual accounts of individuals within their systems or as a part of their contractual obligations– often as part of their legal duties, which points to the existence of "legal means" to identify in these cases.

Wallet providers have a significant role in this debate as they issue and manage WUAs. Moreover,during the activation, the Wallet Provider requests device data from the Wallet Instance, including supported communication technologies and details of the Wallet Secure Cryptographic Device (WSCD), such as an embedded Secure Element, SIM, or e-SIM.[12]. It is, however, not clear from the ARF whether a wallet provider can identify a natural person based on these identifiers. The answer would be in the affirmative if our analysis relied on the pre-SRB period, as these identifiers would be considered pseudonymous data and thus would be subject to the GDPR. The EUDIF framework explicitly allows a single entity to combine multiple roles (e.g., being both a Wallet Provider and a PID Provider or a TSP) as long as they comply with the specific legal and technical requirements for each individual role, PID providers, TSPs and wallet providers can be the same organisations; in such cases, the linkability of datasets is more likely than when they are held by different organisations.

Several privacy-enhancing techniques are applied to the EUDI, including selective disclosure and data minimisation to increase granular control. Technically, ISO/IEC 18013-5 and SD-JWT VC standards enable salted hashes for the attributes, which are used with pseudonyms to improve the unlinkability of datasets across different contexts.[12] Salted hashes have been discussed previously by legal scholar Finck in the context of first-generation blockchains. She argued that hashing – or encryption– in general does not achieve anonymisation – they merely achieve pseudonymisation, thus they are subject to the GDPR. [16]. By the same logic, public keys constitute personal data as they are unique identifiers. This approach tends to rely on an absolute approach to personal data; in the context of EUDIW, this interpretation should be nuanced.

When the SRB case is applied, the focus here should be on who has reasonably likely means to identify the users, which boils down to the question: who holds the key? According to the ARF, private keys are held solely by users within the device's secure hardware. Wallet providers, who can manage the wallet unit, including its recovery, cannot access the keys or the contents within the wallet, i.e., PID and EAA (ARF 6.5.4) While their role is essential for the processing – they determine the purposes and means of the processing as in the definition of data controller – (if) they cannot access critical personal data, which raises questions with regard to who is the data controller for the private keys? As private keys are stored on the device hardware, one might argue that device manufacturers are responsible for processing these sensitive data. One can also argue that, for an entity to be qualified as a controller, access to data is not required[22][23]. This, however, applies only in situations where joint controllership has been established – which is not the main focus of this paper and, due to space restrictions, can only be addressed in further research.

Last but not least, data matching for TSPs in EUDIF is prohibited. Providers are required to keep personal data linked to the wallet logically separate from any other data they hold, preventing the merging of identity data with other data for profiling (article 5(a)14).[5]. This is confusing when considering the DOP amendment of "if the risk of identification is prohibited by law or impossible in practice…"[19]. This might mean that while practical means of identification are possible, but due to the legal prohibition, the data in question cannot be considered personal.

## 5. Concluding Remarks: Clarity, Simplification or a Responsibility Gap?

First, in decentralised identity management systems, the legal implications can be said to be necessarily segmented across multiple actors and data types within the same ecosystem. This analysis shows that while not all data would be considered personal for all entities that are involved in the EUDIW processing, residual linkability persists within this ecosystem. Certain actors, by virtue of their role, process these data in combination with other datasets. This creates a reasonable likelihood of identification. At the same time, assessing "the means reasonably likely to be used" remains exceptionally difficult for outsiders, including regulators and researchers, given the complexity of the data chain. Thus, it can be argued that the complexity and opacity still challenge the practical application of the personal data doctrines.

Accordingly, one can conclude that recent regulatory and case law developments do not necessarily resolve this uncertainty. For example, the processing of a device identifier by a wallet provider results in a unique identifier persistently associated with a specific user identity. However, a wallet provider can claim they have no reasonably likely means to identify the user, while in fact they can single out users using this identifier to maintain their role as a wallet provider. Who is responsible for the said processing is unclear and appears fragmented/obscured, which might leave data subjects without a clearly identifiable, accountable actor against whom to exercise their rights. This fragmentation risks creating responsibility gaps, particularly when technological design choices are driven by an incentive to anonymise data to avoid GDPR rather than to ensure accountability. Future research should therefore examine whether relationships among these entities amount to joint controllership within the meaning of Article 26 of the GDPR, and how they should be allocated.

## Acknowledgments

This research has been funded by the EU NOUS Horizon Project No. 101135927 and the Cybersecurity Research Program Flanders – second cycle, funded by the Flemish Government.

## Declaration on Generative AI

During the preparation of this work, the author used ChatGPT-5.2 in order to learn how to use LaTeX and Overleaf.

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
