# OpenReview forum: "Personal data in EU digital wallets in light of recent developments in EU data protection law"
_SolidProject.org/SoSy/2026/Privacy_Session — SoSy2026-Privacy Paper_

### Official Review · ~Andres_Chomczyk_Penedo1 · 2026-02-25
**Good and suitable paper for the conference**

**Rating:** 8
**Confidence:** 5

**Review:**

-	Quality:
o	The paper is a strong submission for the workshop.
o	Relevant caselaw and literature is adequately referenced.
-	Clarity:
o	While the topic is complex, particularly for non-legal audiences, the author(s) make a very good work in focusing on the main issues under consideration and doing so in a concise manner.
o	The paper is quite clear on (i) its objectives, (ii) its methodology, and (ii) its intended results/outcome, making it scientifically sound.
o	While there is a research question, which much appreciated, there might some typos involved that could be revised to improve readability.
o	In the same vein, it would be beneficial to conduct further proofreading to correct some minor typos, such as missing spaces, and some wording used.
o
-	Originality:
o	The study conducted is novel, particularly thanks to the policy and legislative developments covered in the paper.
o	While some commentaries have emerged tackling the new paradigm laid out by the involved caselaw and legislative proposals, these have been limited to other venues different than an academic publication, such as these. Hence, the novelty factor is present in here.
-	Significance:
o	The paper unpacks a highly relevant topic for the operation of digital wallets in light of the identified debate of what constitutes personal data, triggering the application of GDPR.
o	The future research agenda identified by the author(s) is relevant; however, there have been previous attempts to address, at least partially, this question. Perhaps the limited space allowed in this submission conditioned the works referenced.

---

### Official Review · ~Radina_Stoykova1 · 2026-03-04
**Personal data in EU digital wallets in light of recent developments in EU data protection law**

**Rating:** 7
**Confidence:** 4

**Review:**

see attachment.

---

### Decision · Program_Chairs · 2026-03-09

Accept (Paper)